# Reflections on Recidivism and Relapse Prevention among Italian Justice-Involved Juveniles: A General Overview

Valeria Saladino [1], Danilo Calaresi [2,*], Filippo Petruccelli [3] and Valeria Verrastro [2]

[1]  Department of Human, Social and Health Sciences, University of Cassino and Southern Lazio, 03043 Cassino, Italy; v.saladino@unicas.it
[2]  Department of Health Sciences, University of "Magna Graecia", 88100 Catanzaro, Italy; valeriaverrastro@unicz.it
[3]  Institute for the Study of Psychotherapy, 00185 Roma, Italy; filippopetruccelli@istitutopsicoterapie.it
[*]  Correspondence: danilo.calaresi@unicz.it

**Abstract:** Research interprets antisocial and illegal behavior among juveniles as an expression of needs, as a conscious action, or as an adherence to family, cultural, and social contexts. Professionals and researchers interested in the topic could benefit from reflections and insights on relapse prevention among justice-involved juveniles (JIJs). In light of these considerations, we investigated the criminal conduct of JIJs, identifying their background, individual characteristics, and the educational and rehabilitative programs of the 17 Italian youth detention centers from a sample of 234 JIJs (214 males and 20 females, 14–25 years old). The sample completed the following questionnaires: the high-risk situation checklist, deviant behavior questionnaire (DBQ), and the neighborhood perception questionnaire (NPQ). The study aims to provide a general overview of the justice-involved adolescents and young adults in Italian youth detention centers, focusing on perpetrator profiles, family systems and the quality of life in the Italian youth detention centers. To achieve our goals, we investigated their occupations and education, the perceived quality of life in their neighborhoods, the use of drugs, and the tendency to commit illegal or antisocial behaviors before incarceration. The study also explored the awareness related to the personal perception of the risk factors in relapse, with the aim of stimulating reflections on behavior and crime-related cognitions to promote relapse prevention. We discuss the main findings and future implications.

**Keywords:** recidivism; relapse prevention; justice-involved juveniles; rehabilitative programs; Italian youth detention centers

## 1. Introduction

### 1.1. Recidivism among Justice-Involved Juveniles

Research on recidivism aims to investigate the developmental process that impacts youth criminal behavior, to identify the main relapse-related risk factors. Studies from Italy (Guarnaccia et al. 2022) and the Netherlands (Mulder et al. 2019) have identified common factors linked to increased recidivism risk. High-risk juveniles typically exhibit a spectrum of issues, encompassing physical and/or mental disorders, low self-esteem, poor social skills, deficient problem-solving and coping strategies, and a history of violent behavior (Saladino et al. 2021a). Additionally, they may have a family history of alcoholism, involvement in substance abuse and/or deviant peer groups, and a history of scholastic drop-out (Saladino et al. 2020; Verrastro et al. 2024). Mulder et al. (2019) emphasize genetic predisposition, learning and neurodevelopmental issues, and negligent or severe parenting as contributing factors (Saladino et al. 2024a). Other recidivism risk factors in youths encompass specific personality traits and previous violence (Navarro-Pérez et al. 2020).

Estimating the recidivism rate, particularly among juveniles, is challenging. For instance, Italian research faces efforts to provide reliable data due to a lack of monitoring of individuals released from correctional facilities (Guarnaccia et al. 2022). Researchers

often use tools like the Structured Assessment of Violence Risk in Youth and Youth Level of Service/Case Management Inventory 2.0 (SAVRY and YLS/CMI) to predict recidivism risk in young offenders (Ortega-Campos et al. 2020a, 2020b). In the study of recidivism among youths, protective factors have also been considered, such as those factors and/or resources that have a positive impact by reducing recidivism risk. In this line, research highlights the following as the common protective factors in relapse prevention: the presence of family and community support that provide emotional and social closeness; access to quality education that offers learning opportunities for personal development (Navarro-Pérez et al. 2020); social engagement, such as attending extracurricular activities, both in sport or within the community, to establish positive relationships within their peer group (Guarnaccia et al. 2022); the presence of positive models which could inspire juveniles to avoid illegal or dangerous behaviors, and the possibility to learn coping strategies to manage negative feelings, often related to criminal decision-making (Wang et al. 2018).

*1.2. From Nothing Works to What Works*

The reduction of the recidivism rate represents an indicator of success in re-educational interventions for juveniles, as highlighted by the Observatory and Data Bank on the Phenomenon of Juvenile Deviance in Europe (Mastropasqua et al. 2013). The effectiveness of rehabilitation interventions for formerly incarcerated individuals has long been a subject of scientific debate. In 1974, sociologist Robert Martinson (1974) stirred the debate with his article "*What works: Questions and answers about prison reform*", summarizing 231 studies from 1945 to 1967. Initially, Martinson asserted that most rehabilitative efforts showed no appreciable effect on recidivism (Riley 2011, p. 139), spawning the "nothing works" approach. However, Martinson later revised this stance, acknowledging instances where treatment and rehabilitation were effective, but this perspective struggled to resonate within the social and scientific communities. Concurrently, Lamar Empey (Smith et al. 2009), a University of Southern California sociologist, emphasized the critical role of the treatment delivery conditions in achieving success. Empey argued that rehabilitation's efficacy hinges more on creating favorable conditions, than on specific treatment methods. The "Provo Project" exemplified this concept, revealing significantly lower recidivism rates for youth offenders in community-based programs compared to those in state institutions (Ibid.).

In 1987, Gendreau and Ross (1987) presented a positive perspective on treatment effectiveness by analyzing over 200 studies on rehabilitation from 1981–1987. Their findings, indicating successful treatments in both community and institutional settings, demonstrated an 80% reduction in recidivism over a two-year follow-up period (Riley 2011). Several intervention and treatment strategies from this survey encompass family therapy, cognitive problem-solving, support for independent living, on-the-street monitoring, negotiation skills, modeling, interpersonal skills, behavior contracting, individual and group therapy, and job training. The subsequent proliferation of evaluations of offender programs in professional journals fueled the contentious "what works" debate, characterized by heterogeneous results due to diverse populations, criteria of success, and methodologies. As a solution to this complexity, Mark Lipsey analyzed 450 correctional outcome studies. Lipsey concluded that treatments yielded approximately a 10% reduction in recidivism (Ibid.). However, subsequent meta-analyses, such as by Zinger et al. (Ibid.), provided varied insights, emphasizing that targeted approaches to higher-risk offenders achieved positive outcomes. This perspective laid the foundation of "risk" and tailoring treatments to specific target populations (Marshall and McGuire 2003).

On the same line, Bonta and Andrews (2017) formulated the principle of balance between the risk and the treatment, describing the concept of criminogenic and non-criminogenic needs. Criminogenic needs are dynamic factors because they change over time, influencing recidivism, and are part of the overall risk level of the offender. Non-criminogenic needs are also dynamic factors, but are not associated with crime and recidivism. Finally, static factors, such as family history and background, influence the possibility of reoffending, but are stable during the life course. Another principle identified by the

authors (Ibid.) refers to the responsivity of the offenders to the interventions. According to this principle, the treatment should be appropriate for the capabilities of the offender, considering their social and family background, intelligence, and cognitive and verbal ability. Thus, rehabilitation programs for offenders should evaluate risk/need and responsivity principles in their assessment.

This scientific debate increased knowledge about the efficacy of intervention in rehabilitation among formally incarcerated people, stimulating future directions in relapse prevention programs.

### 1.3. The Efficacy of Social Rehabilitation in Correctional Facilities

To date, the efficacy of treatment in correctional facilities is evaluated by considering the different category of offender, the treatment approach, and the characteristics of the correctional institutions, according to the following phases: assessment, treatment design, and treatment management. During the assessment phase, the risk of relapse is gauged along with criminogenic needs, guiding the treatment's intensity. Precision in intervention planning has improved, notably with tools like the Level of Service Inventory–Revised and the Offender Assessment System for Adults, and the Youth Level of Service/Case Management Inventory for Youths (Cigno and Bourn 2017; Howard 2006). These tools provide a comprehensive risk/need score, encompassing family circumstances, parenting, education, employment, peer relations, substance abuse, leisure time, personality, and attitudes. Asset, adapted by the Youth Justice Board for England and Wales, is another assessment tool with practitioner and self-report sections. These tools, informed by research literature on recidivism risk factors, assess and weigh various elements to determine an individual's risk score. The YLS-CMI and Asset demonstrate predictive efficacy in general recidivism for young offenders (Olver et al. 2009).

The treatment design phase prioritizes relapse prevention through structured approaches and fostering changes in cognition and behavior. Research underscores the greater efficacy of focused and structured treatments over general ones. Cognitive, behavioral, and family therapies prove more beneficial for general offender samples compared to traditional psychodynamic and nondirective client-centered therapies. The primary goal is to modify attitudes, values, and beliefs supporting antisocial behavior, promoting prosocial behaviors, and cultivating social and cognitive skills, responsibility, and empathy. Tailoring treatment designs to the individual's personal, cultural, and social background, considering intelligence, verbal, and cognitive skills, enhances effectiveness when evaluated within the risk/need and responsivity principles (Cigno and Bourn 2017).

Finally, the treatment management phase regards the application of the intervention according to the previous two phases. A coherent intervention in intentions and goals has a high level of integrity and promotes positive outcomes. This coherence produces more quality and control in managing and improving the program (Ibid.).

### 1.4. Italian Youth Detention Centers

In the Italian context, the Social Service Offices for Minors (USSM) intervene in every level of the criminal proceedings, from the first phase until the conclusion of the judicial process. According to the Italian judicial system, an individual can be tried and convicted from the age of 14 onwards; based on the same system, adolescents, and young adults (14–25 years of age) are taken into charge by the youth detention center (Decree-Law No. 92 of 2014—Interventions in Penitentiary Matters 2014). The following social services for JI juveniles are involved in the process:

- First reception centers (Centro di Prima Accoglienza-CPA): These centers host minors for up to 96 h. The judge (Giudice per le Indagini Preliminari GIP) assesses the arrest validity during validation and decides on one of four precautionary measures for minors;
- Therapeutic communities (Comunità terapeutiche): These ministerial or private entities host minors under precautionary measures outlined in Presidential Decree 448/88

(placement in the community). Communities may serve as probation or alternative security measures;

- Youth detention centers (Istituto Penale Minorile—IPM): These institutes are aimed at pre-trial detention and imprisonment. IPM provides adequate responses to youth users and the requirements connected with the judicial authority provisions. A multidisciplinary team manages the treatment activity. A stable socio-educational reference operator belongs to the administration, so training, professional, cultural, and animation activities are provided in collaboration with operators of other educational professions, such as private social and voluntary associations.

According to the report of the Ministry of Justice (Zanghi et al. 2018), the Juvenile Criminal Procedure Code—Presidential Decree 448/1988 (1989), and the Legislative Decree No. 121 of 2 October 2018—Regulations for the Execution of Sentences against Convicted Minors (GU.26.10.2018) (2018), titled "Discipline of the Execution of Sentences Against Convicted Minors", in recent years, society has witnessed an increasing application of community placement and other alternative judicial measures, to avoid detention and promote educational needs.

*1.5. Aim of the Study*

Drawing from the literature on risk factors associated with recidivism among justice-involved juveniles, and given the social impact of the topic, our aim is to examine recidivism risk factors. Specifically, our study presents data on the personal perception of relapse among justice-involved adolescents and young adults in Italian correctional facilities, providing insights into primary risk factors related to crime and recidivism. We delve into this perception to prompt reflections on relapse prevention, highlighting social rehabilitation as a central focus within the legal and educational system.

## 2. Materials and Methods

*2.1. Participants and Procedures*

Our sample consists of 234 adolescents and young adults, 214 males and 20 females, in Italian youth detention centers (*n* = 17), with an average age of 18.90 (SD = 2.21; range of age 14–25). The procedure is applicable to foreign nationals who are fluent in the Italian language and who are accompanied by a cultural mediator. The protocol was assessed and authorized by the Ministry of Justice and the directors of the participating youth detention centers. The protocol was carried out during the hours allocated to laboratory and school-related activities, under the supervision of teachers, educators, the research team, and police officers. Participants were assured of the complete anonymity of their data. The study was conducted following the Declaration of Helsinki and approved on 9 October 2019, by the Institutional Review Board of the University of Cassino and Southern Lazio.

*2.2. Measures*

The sociodemographic questionnaire, developed by the researchers of the University of Cassino and Southern Lazio, investigates the main risk factors in deviant behavior, such as history of substance abuse, family issues, neighborhood environments, perceived discipline, recidivism, and general level of well-being.

The deviant behavior questionnaire (DBQ), from the International Self-Reported Delinquency Study (Enzmann et al. 2015), adapted in Italian by researchers from the University of Cassino and Southern Lazio (Saladino et al. 2020), is composed of nine items which investigate risky and antisocial behaviors during adolescence, evaluated dichotomously. The DBQ includes items on the tendency to commit illegal activities ("Have you ever stolen something from a store?"; "Have you ever entered a shop with the intent to steal?"; "Have you ever stolen a bicycle, a scooter or a car?"; " Have you ever illegally downloaded music or movies from the internet?"), and items on the tendency to aggressive attitudes ("Have you ever threatened or assaulted someone with a weapon to steal their money or their belongings?"; "Have you ever attacked someone verbally or physically?"; "Have you ever

intentionally damaged something, like a bus shelter, a window, a car or a place in the bus or train?"; "Have you ever carried a weapon with you such as a stick, a knife or a chain?"; "Have you ever participated in a brawl, for example at school, at the stadium or in a public place?").

The neighborhood perception questionnaire (NPQ), developed by the research team of the University of Cassino and Southern Lazio (Ibid.), is composed of 11 items and investigates the personal perception of the neighborhood, analyzing the sense of safety ("I felt safe in my neighborhood"; "I often witness crime") in terms of satisfaction ("I felt satisfied with the activities and services offered"), and the general tendency for sociality ("When I had a problem, I could ask the neighbors for help"). The items are evaluated using a Likert scale (1 = totally false; 5 = totally true).

The high-risk situation checklist, developed by David M. Price (1999) and adapted in Italian by researchers from the University of Cassino and Southern Lazio (Saladino et al. 2020, 2023), evaluates the perception of the risk related to recidivism, underlining emotional, social, situational, and possible treatment variables. It consists of 63 items. In this study, specifically, it evaluates the factors connected to the hypothetical possibility of relapse, including negative emotions, positive emotions, thoughts and actions, the characteristics of the environment, rehabilitation, and other positive or negative situations that could lead to (a) a person who never committed a crime to do so, and (b) a person already involved in the justice system, to relapse. It is important for clinicians and psychologists to evaluate and establish possible strategies to prevent problematic behaviors and to promote social rehabilitation of justice-involved juveniles during and after detention experiences.

### 2.3. Statistical Analysis

Data were analyzed using the Statistical Package for Social Sciences (Version 26.0, SPSS Inc., Armonk) (IBM Corp 2019, released). To account for the influence of background variables, a multivariate analysis of covariance (MANCOVA) was employed. The MANCOVA included deviant behavior and recidivism as dependent variables, gender and nationality were treated as fixed factors, while age served as a covariate. In cases where the background variables exhibited significant multivariate impacts, further univariate analyses, as well as pairwise comparisons (Bonferroni correction), were executed.

Furthermore, correlational analyses were carried out to explore the relationships between recidivism, deviant behavior, and neighborhood perception.

### 3. Results

#### 3.1. Risk Factors

Our sample is composed of Italian (65.8%) males (91.5%), aged between 14–25 years old (66.2%), from the south of Italy (48.3%), who were mostly involved in property crime (47.4%).

Regarding the percentage of recidivism among the sample, 19.7% were at the first detention, 20.9% were recidivists, and the remaining 59.4% represents cases of not declared data, due to privacy reasons related to some correctional facilities negating the authorization to collect this information. When analyzing the sociodemographic questionnaire, some factors emerged that could influence recidivism rate, confirming the main literature on the topic (Table 1).

**Table 1.** Risk factors in juvenile recidivism.

| Risk Factors | % |
|---|---|
| Poor education | 74.8 |
| Substance abuse | 65.8 |
| Perceived severe discipline in childhood | 53.8 |
| Absence of school/work activities | 47.9 |
| Justice-involved parent/s | 31.2 |
| Psychopharmaceutical drugs without prescription | 16 |

### 3.2. Deviant Behavior Tendency

Concerning the tendency to behave illegally or antisocially before incarceration, evaluated by the deviant behavior questionnaire (DBQ), the most common behavior among adolescents was aggression and fighting (Table 2).

**Table 2.** Perpetrators' illegal or antisocial behaviors before incarceration.

| Behaviors | N | % |
| --- | --- | --- |
| Fighting | 156 | 66.7 |
| Aggression | 136 | 58.1 |
| Vehicular theft | 122 | 52.1 |
| Illegal possession of weapons | 121 | 51.7 |
| Cyberspace illegal acts | 119 | 50.9 |
| Stealing (in a store) | 113 | 48.3 |
| Damage to public property | 104 | 44.4 |
| Intent to steal | 83 | 35.5 |
| Threats-with weapon | 69 | 29.5 |

### 3.3. Neighborhood Perception

The evaluation by the neighborhood perception questionnaire (NPQ) showed that 54.3% of the participants felt safe in their own neighborhood and 38% of them affirmed that they did not witness a crime; while 29.9% had a low sense of satisfaction about their neighborhood services and support, and only 12.8% perceived a sense of sharing and closeness from neighbors. Despite the negative feelings about the lack of opportunities and the sense of solitude in their neighborhood, 59.8% of the participants affirmed that they liked living there, and 48.3% said that they wanted to return there after their release from the youth detention center.

### 3.4. High-Risk Situation and Perception of Recidivism

An important aspect analyzed was the personal evaluation of the factors which could increase the risk of recidivism. Data from the checklist of high-risk situations showed the main characteristics which influence recidivism, according to youths involved in justice. The checklist is structured according to six categories and participants were asked to reflect on the impact of the listed factors in relapse: negative emotions, positive emotions, thoughts and behaviors about crime, neighborhood characteristics, and feeling about rehabilitation programs, and other positive or negative situations. This evaluation is based on self-regulation, self-image, and consciousness of the crime. The high-risk situation checklist proposes an imaginative and hypothetical exercise to evaluate future conduct and to investigate self-criticism and recognition of personal strengths and fears about the offence. The results are a valid starting point to creating social rehabilitation programs, based on offenders' awareness, resources, and critical issues. However, it is difficult for JI youths to think about these themes, because on one hand, they feel that they would not commit another offence, being aware of the consequences, and on the other hand, they do not trust the society. Consequently, after the release from the center, they are afraid, lonely, and resigned about the lack future perspectives. Thus, those who have beliefs in themselves and in their abilities, and who have family or friend support, are more likely to interact positively with their environment, avoiding relapse.

Negative emotions: The most common negative emotion associated with relapse is rage and problems in managing it (Table 3). Indeed, negative feelings associated with rage are commonly related with a higher risk of behaving in aggressive and impulsive manners.

**Table 3.** Negative emotions and recidivism.

| Item | Percentage |
|---|---|
| Rage and problem management | 47.9 |
| Anxiety | 15.0 |
| Frustration | 14.5 |
| Solitude | 12.8 |
| Depression | 12.4 |
| Rejection | 12.4 |
| Resentment against others | 11.5 |
| Grief | 11.1 |
| Blame | 10.7 |
| Shame | 9.4 |
| Unrealistic fears | 6 |
| Sense of inadequacy | 4.3 |
| Self-pity | 3.8 |

Positive emotions: The most common positive emotion associated with relapse is the overconfidence in avoiding other crimes. Excessive confidence, often associated with the sense of omnipotence and grandiosity, widespread among young people, seems to correlate with a lower propensity to reflect on oneself, to consider caregiver suggestions, and to allow themselves to be influenced in criminal actions. Therefore, a positive emotion, such as security regarding one's ability to no longer commit offences, if excessive, might become risky, since it prevents one from reflecting on one's actions, and leads to impulsive decisions (Table 4).

**Table 4.** Positive emotions and recidivism.

| Item | Percentage |
|---|---|
| Sense of control | 41 |
| Overconfidence in avoiding other crimes | 34.2 |
| Relief from physical and emotional tension | 13.2 |
| Think <<I will not commit other offences>> | 12.4 |
| Magnificence and omnipotence | 10.3 |

Thoughts and behaviors about crime: Among thoughts and behaviors that could affect relapse, the participants identified the item "Thinking <<I will not do it anymore>>" (Table 5), as the most impactful.

**Table 5.** Thoughts and behaviors about crime and recidivism.

| Item | Percentage |
|---|---|
| "Thinking <<I will not do it anymore>>" | 29.9 |
| My behavior is correct | 23.5 |
| I cannot do anything else | 17.1 |
| I cannot control my behavior | 16.2 |
| I do not know the consequences of my actions | 11.5 |
| The consumption of drug and alcohol | 11.1 |
| Believing I will not commit other crimes | 10.7 |
| My life has no meaning | 10.3 |
| I do not trust psychological programs of rehabilitation | 9.4 |
| Thinking that I am different from others | 9.4 |
| I think of crime continuously | 9 |
| I have fantasies, dreams, and thoughts about crimes | 8.1 |
| I cannot manage stress | 7.7 |
| Wanting to give a false impression to others | 5.6 |
| I feel better if I am involved in criminality | 2.1 |

Neighborhood characteristics: The most diffused characteristics of the neighborhood which could affect the risk of recidivism is the easy access to weapons, a characteristic that describes a criminogenic environment and that could be also associated with the easy access to drugs and alcohol, and the contact with other people involved in criminality (Table 6). Indeed, all these aspects are often related to, and common in deviant peer groups, gangs, or criminal organizations.

**Table 6.** Neighborhood characteristics and recidivism.

| Item | Percentage |
|---|---|
| Easy access to weapons | 27.8 |
| Easy access to drugs and alcohol | 24.4 |
| Contact with other people involved in criminality | 23.1 |
| Involvement in criminal actions | 20.5 |
| Poor social skills | 12.4 |
| Interpersonal conflicts | 11.1 |
| Presence of potential victims | 10.7 |
| Conflict with partner/friends | 7.7 |
| Social isolation and solitude | 6 |

Feelings about rehabilitation programs: One of the most common feelings about rehabilitation programs offered after release is the difficulty to trust and adhere to it. The participants reported that not participating in rehabilitation programs could lead to relapse (Table 7). This response is in line with the attitude commonly shown by JIJs, who in the context of correctional facilities do not show trust in the institution and in the operators of the treatment area, focusing more on concrete and material aspects, such as economic well-being, rather than the psychological ones, not asking for psychological support, but rather to receive a work tasks to compensate for the economic issues of the family system or background.

**Table 7.** Rehabilitation programs and recidivism.

| Item | Percentage |
|---|---|
| Not participating in rehabilitation programs | 34.6 |
| Difficulty in trusting others | 32.1 |
| No trust in programs | 19.2 |
| Incapability to be honest with the professionals | 16.2 |
| Not following the advice given by the professionals | 16.2 |
| Non-awareness of the situations which could result in the commitment of other crimes | 9 |
| Incapability to be involved in support group | 6.8 |
| Lack of information about the offence and its consequence | 4.3 |

Other situations: Among other situations, positive or negative, which can increase or decrease the possibility of committing another crime, the participants reported that success at work is the most important factor (Table 8), consistent with the idea that economic well-being is the only solution to solve their problems.

**Table 8.** Other situations and recidivism.

| Item | Percentage |
|---|---|
| Work success | 48.3 |
| Thinking about the future | 40.6 |
| Use of leisure time | 15.4 |
| Controlling the desire to commit other crimes | 12.8 |
| Avoid expressing my own feelings | 12 |
| Incapacity to talk about the desire to commit a crime | 4.7 |

### 3.5. Italian Youth Detention Centers

Participation in the research was on a voluntary basis, therefore the percentage reported in this paragraph does not involve all the juveniles allocated in the 17 Italian Centers but only those who decided to adhere to the study. Data was aggregated according to the geographical distribution; for privacy reason research avoids showing youth detention center location or name. Most of the sample come from the South of Italy and the distribution of the perpetrators agrees with their provenience. Also, there is a high percentage of foreigners in youth detention centers of the south and of the center of Italy. Regarding the treatment activities in the Centers, 54.7% of the participants are involved in scholastic programs and 72.2% of them are involved in work or recreative activities, both positive data, while only 12.8% of the sample received a psychological support in the Center. Mostly, participants are involved in gardening and restaurant activities, as reported in Table 9.

**Table 9.** Types of professional and recreational activities in youth detention centers.

| South and Islands Youth Detention Centers | |
|---|---|
| Center 1 | Restaurants, gardening, masonry |
| Center 2 | Restaurants, barber activity, masonry |
| Center 3 | Ceramics, restaurants, sports, cleaning |
| Center 4 | Maintenance, cleaning, masonry, crib art, ceramics, restaurants, gardening |
| Center 5 | Locksmith activity, masonry, gardening |
| Center 6 | Restaurants, gardening, masonry |
| Center 7 | Restaurants, gardening, masonry |
| Center 8 | Restaurants, cleaning, masonry |
| Center 9 | Cleaning, masonry, and volunteer activities |
| Center 10 | Carpentry, gardening, cleaning |
| **Center Youth Detention Centers** | |
| Center 11 | Artistic drawing, restaurants, cleaning |
| Center 12 | Music, art drawing and theater labs, restaurants, cleaning, masonry, sport activities, gym, gardening |
| Center 13 | Librarian, tailoring, sport activities, masonry, cleaning |
| **North Youth Detention Centers** | |
| Center 14 | Restaurants, electrician, cleaning, art labs, typography |
| Center 15 | Restaurants |
| Center 16 | Gym |
| Center 17 | Theater lab |

### 3.6. Descriptive and Correlational Analysis

Correlational analyses are shown in Table 10. Further analyses were carried out to assess the impact of gender, nationality, and age on study variables. However, no significant multivariate effects were found, indicating that these variables did not have a substantial impact on the variables under investigation (gender: Wilks' $\lambda = 0.99$, $F[2, 87] = 0.20$, $p = 0.82$, $\eta p^2 = 0.01$; nationality: Wilks' $\lambda = 0.99$, $F[2, 87] = 0.14$, $p = 0.87$, $\eta p^2 = 0.01$; age: Wilks' $\lambda = 0.98$, $F[2, 87] = 0.82$, $p = 0.45$, $\eta p^2 = 0.02$).

**Table 10.** Correlations between eecidivism, deviant behavior, neighborhood perception.

| | 1 | 2 |
|---|---|---|
| 1. Recidivism | - | - |
| 2. Deviant Behavior | 0.32 ** | - |
| 3. Neighborhood Perception 3—Had many friends | −0.02 | 0.02 |
| 4. Neighborhood Perception 4—Spent free time outdoors | 0.16 | 0.37 ** |
| 5. Neighborhood Perception 5—Spent free time at home | −0.10 | −0.29 ** |
| 6. Neighborhood Perception 6—Felt secure | −0.08 | −0.06 |

**Table 10.** *Cont.*

|  | 1 | 2 |
| --- | --- | --- |
| 7. Neighborhood Perception 7—Often bored | −0.01 | −0.07 |
| 8. Neighborhood Perception 8—Witnessed crimes | 0.40 ** | 0.40 ** |
| 9. Neighborhood Perception 9—Being able to ask neighbors for help | −0.06 | −0.18 * |
| 10. Neighborhood Perception 10—Satisfied with activities/services | −0.24 * | −0.13 |
| 11. Neighborhood Perception 11—Liked the neighborhood | 0.08 | 0.13 * |
| 12. Neighborhood Perception 12—Want to stay there | 0.06 | 0.05 |
| 13. Neighborhood Perception 13—Graffiti and abandoned buildings | 0.22 ** | 0.18 * |

Note: * $p < 0.05$. ** $p < 0.01$.

## 4. Discussion

*Main Findings*

A total of 234 questionnaires were administered to the participants in the Italian Youth Detention Centers ($n = 17$). The sample was composed of males and females, adolescents, and young adults. Most of our sample (47.4%) had engaged in property crimes, while 24.8% had been involved in violent offenses. For privacy reasons, 13.7% of offenses and 59.4% of the recidivism rate were not declared from the direction of correctional facilities. Examining justice-involved juveniles' background revealed some significant factors such as low education, unemployment, substance abuse (particularly cannabis and cocaine), and unsupervised use of psychopharmaceutical drugs to address psychological issues, that according to the literature, are often related to higher risk behaviors (Saladino et al. 2020; Lösel and Farrington 2012; Jolliffe et al. 2017). The use of substances should be monitored to prevent young offenders from using them as a coping strategy to face stressful events (Saladino et al. 2020, 2024b). Analysis of antisocial behaviors before incarceration, using the deviant behavior questionnaire (DBQ), highlighted a prevalent inclination toward aggression and fights among the sample. This result suggests the use of violence as a primary means of communication. Adolescents and young adults in the justice system perceive criminal activities as essential for achieving goals and constructing their identity. Moreover, exiting a challenging situation is complicated due to family system involvement and the influence of a crime-oriented education (Saladino et al. 2020).

Data from the neighborhood perception questionnaire (NPQ) showed that 54.3% of the sample felt safe in the neighborhood. The sense of satisfaction about the services offered (29.9%), as well as the sense of sharing and helping from neighbors, was low (12.8%). Despite the negative feelings about the lack of opportunities and the sense of solitude, participants affirmed that they like living there (59.8%), expressing the will to continue to live in the same neighborhood after release (48.3%). At the same time, data from the high-risk situations checklist show that according to the participants, the most prevalent factors for risk for recidivism related to the environment are the presence of weapons (27.8%) and contact with other people involved in criminality (23.1%). They seem to perceive their neighborhood as safe; at the same time, they are aware of the strong possibility of being involved in criminal activities and having easy access to weapons and drugs. Correlational analyses between deviant behavior, recidivism, and neighborhood perception confirmed the mentioned considerations, showing a positive relationship between deviant attitudes and recidivism, and some environmental characteristics such as witnessed crimes and neighborhood degradation, potentially due to the general tendency to develop a criminal career and use offenses as a problem-solving strategy. A negative correlation emerges between deviant behavior and the sense of sharing and closeness from neighbors, possibly underlying the protective role of closeness and disclosure within the neighborhood's members in criminal decision-making. Another aspect that seems to be related to a higher risk of deviant behavior is the tendency to spend free time outdoors, probably because of the higher exposure to a potential criminogenic environment, the contact with a deviant peer group, and the absence of parental control (Saladino et al. 2020).

The environment has an impact in terms of behavioral influence. Justice-involved juveniles are often involved in crimes associated with their neighborhood; they commonly witness crimes or are involved in deviant actions or deviant groups—for this reason, we can interpret their decision not to return to their neighborhood after release as a protective factor, both in terms of awareness and with regard to the actual involvement of the minor in crimes. The perception of one's neighborhood and the decision to return after release are influenced by internalized norms and the presence of alternatives. Mostly, adolescents and young adults identify with conformity to the norm as the only way to achieve their goals and build their own identity, and in many contexts, the norm is equivalent to a deviant background and career (Brinthaupt and Scheier 2022). Furthermore, juveniles who received education based on the culture of violence commonly do not have the possibility to detach from their family and environment (Morton 2022). In this frame, formally incarcerated adolescents perceive correctional facilities as a rite of passage and as a part of their idea of personal growth (Bajari and Kuswarno 2020).

In examining the family context, 53.8% of the sample reported experiencing severe discipline during childhood, aligning with research indicating widespread parental abuse or neglect among justice-involved juveniles (Saladino et al. 2024a). This association is further supported by studies highlighting the moderating role of parental discipline styles in anger and aggression among juvenile offenders (Tavassolie et al. 2016). The sample reflects a high unemployment rate among parents, particularly mothers. While 73.2% of parents are married, the majority have a junior high school education. Additionally, 31.2% of fathers have a history of criminal records, and 9.2% are drug users, contributing to a background characterized by risk factors associated with lower employment, education, and risky behaviors, particularly among fathers. Literature underscores the influential role of these factors in delinquent and deviant behavior among juveniles, with a specific emphasis on the significant impact of fathers, particularly for justice-involved boys (Tapia et al. 2018).

Regarding the personal perception of high-risk situations in relapse, the high-risk situation checklist revealed the characteristics that could lead those involved in crime to relapse, based on evaluation grounded on six categories, as follows: negative emotions, positive emotions, thoughts and behaviors about crime, neighborhood characteristics, and feelings about rehabilitation programs and other positive or negative situations. Among negative emotions, participants identified rage and problems in managing it (47.9%). This response denotes an awareness of this inability, which can be modulated starting from the will. According to this response, our sample recognized that they are not able to manage negative emotions and that these can easily lead to hostile and impulsive attitudes, evolving into real crimes. Especially when the young person does not know other ways of reacting to events, this element could become a risk factor to be worked on at a preventive and rehabilitative level. The failure to manage negative emotions, such as anger, is a factor often associated with aggressive, impulsive, and deviant conduct (Din and Ahmad 2021). Among the positive emotions, juveniles chose a sense of overconfidence in avoiding other crimes (34.2%). Positive emotion means the "positive" evaluation of an emotion that can be translated into a criminal or potentially risky action. For example, if an adolescent thinks he/she is sure to not commit a crime, this feeling could lead to underestimating the risks and committing impulsive actions, incrementing the sense of omnipotence and grandiosity, which is common among juveniles with conduct issues (Fanti et al. 2018). Moreover, the excessive sense of confidence could increment mechanisms of justification and minimization, often related to criminal decision-making (Calvete 2008). The overconfidence in avoiding other crimes coincides with thoughts and behaviors that could affect recidivism, identified on the item "Thinking <I will not do it anymore>" (29.9%). The sense of control over behavior is a typical characteristic of adolescence, a moment in which fear and a sense of omnipotence converge, which often blocks or amplifies certain impulsive actions (Ensink and Normandin 2023). Furthermore, during adolescence, individuals feel the need to assert their autonomy, detaching from adults and

often developing a sense of grandeur that allows them to perceive themselves as more secure (Garrod and Kilkenny 2022). When this sense of grandiosity and security clouds one's ability to reason, it can lead to making dangerous or impulsive choices.

In the environmental characteristics category, the most prevalent element affecting the risk of recidivism is easy access to weapons (27.8%), indicating a link to the criminal environment, together with perceived easy access to drugs and alcohol (24.4%) and contact with other people involved in criminality (23.1%). These aspects could be associated with deviant peer groups, gangs, or criminal organizations (Shapiro et al. 2010; Wojciechowski 2018). Despite this, a low percentage of participants perceive the neighborhood as dangerous, showing an incongruence that reveals a disparity between actual environmental conditions and perception, likely influenced by a desensitization mechanism (Zhang et al. 2021).

Among the attitudes about rehabilitation programs, emerged the difficulty in trusting treatment operators and the absence of compliance towards relapse prevention programs (34.6%). This attitude could be associated with poor awareness of the importance of psychological support in preventing illegal and risky behaviors. Finally, among other situations identified by participants as factors of influence in relapse, work success emerged as the most common element in reducing the risk of recidivism (48.3%). This response denotes a concrete social need that concerns economic well-being related to the quality of life.

Concerning the exploration of the youth detention centers, most of the JI youths come from the south of Italy and the distribution of the perpetrators agrees with their provenience. Moreover, in the sample, there is a high percentage of foreigners in the central and the southern youth detention centers. Half of the participants are involved in educational programs and more than half in work and recreational activities, such as maintenance, cleaning, masonry, art labs, ceramics, restaurants, and gardening, while the rate of psychological support among the centers is low. This aspect could be considered a risk factor for recidivism. Indeed, psychological support within penal institutions, for both adults and minors, aims to increase the critical revision of the crime, improving reflective skills and awareness concerning the elements that contributed to the deviant conduct. This work has, as its primary objective, social rehabilitation and the prevention of recidivism, demonstrating the founding principle of the penitentiary system provided by the Italian law (Presidential Decree 448/1988 (1989) and Legislative Decree No. 121 of 2 October 2018—Regulations for the Execution of Sentences against Convicted Minors (GU.26.10.2018) (2018)). Considering well-known correlations between self-awareness, emotional regulation, and the reduction of impulsivity and aggression (Eadeh et al. 2021), we hypothesize that higher psychological support, aimed at critically re-viewing one's deviant behavior, could increase emotional regulation and constructive reactions, instead of destructive ones, to stressful events (Docherty et al. 2022). Despite the positive emphasis and promotion of the Italian law on psychological support and its role in rehabilitation, the lack of cultural awareness, which is based on violence rather than communication among justice-involved juveniles, and the overcrowding of facilities, could negatively affect rehabilitation programs (Ravena 2019). Moreover, the lack of after-released projects in the Italian context is one of the most dangerous and problematic issues to consider in the evaluation of the risk of relapse in young people.

## 5. Conclusions
### 5.1. Limitations

Our study had the following limits: (1) Because it is cross-sectional, it is difficult to determine which direction is causative, and validating these findings will require longitudinal studies. (2) The study's dependence on self-reported data alone raised the risk of interpretative bias. Consequently, to improve the accuracy of the results, future research projects should strive to include a variety of data sources. (3) The poor sample size reduced the generalization of the findings, so future research should involve larger and more stratified samples.

*5.2. Future Implications*

The present study aimed to provide a general picture and a description of the main characteristics associated with juvenile criminality. It also evaluated the condition of youths in Italian youth detention centers, highlighting some environmental, relational, and individual characteristics that could impact recidivism. Cultural and social models, peer groups, and family systems can influence aggressive and transgressive behaviors, becoming a risk factor in criminal development. Other risk factors could be family with relationship problems and criminal records, risky environment, lack of communication, and poor support.

For this reason, future directions in research and interventions should be based on a multidimensional level. For instance, the so-called family-based approach to therapy, such as multidimensional family therapy (Liddle et al. 2011), is based on the implementation of positive interaction among family members. Also, brief strategic family therapy supports personal skills and new coping capacities among juveniles with behavioral issues (Saladino et al. 2021b).

Moreover, significant contact with family members during the institutionalization process decreases the possibility of relapse after release and increases the success of the re-integration process, based on feelings of understanding and forgiveness, both of which are essential for social rehabilitation (Bahr and Hoffmann 2010). These results suggest the importance of involving family, friends, and other social institutions in rehabilitation programs (Liddle et al. 2011).

Moreover, to promote social and personal transformation in juveniles, it may be useful to adopt a new perspective. An example of this is the restorative justice system, which guarantees a new way to interpret the justice-involved individual and the victim. Restorative justice (RJ) allows reconciliation, avoiding punishment, guided by a facilitator who mediates the relationship between the two parts. RJ is an alternative to vindictive justice and leads the protagonists of the story to regain their roles, to be able to overcome their challenges (McCold 2008). Despite the increased interest in RJ in Italy, its use is limited and occasional. However, a restorative justice paradigm could be consistent with educational purposes, promoting processual outcomes that do not result in "socially stigmatizing" youths.

In light of these considerations, our results and reflections on possible treatments and interventions aim to respond adequately to the different facets of adolescent behavior, to prevent antisocial behaviors, psychopathology, and criminality among juveniles, and to guarantee the social rehabilitation mission promoted by the Italian juridical system.

**Author Contributions:** Conceptualization, V.S.; methodology, V.S. and V.V; data curation, F.P.; writing—original draft preparation, V.S. and D.C.; writing—review and editing, V.S. and D.C.; visualization, F.P.; supervision, V.V.; project administration, V.V.; funding acquisition, V.V. and F.P. All authors have read and agreed to the published version of the manuscript.

**Funding:** The research benefits from the Department of Human, Social and Health Sciences University of Cassino and Southern Lazio funding.

**Institutional Review Board Statement:** The study was conducted in accordance with the Declaration of Helsinki, and approved on 9 October 2019, by the Institutional Review Board of the University of Cassino and Southern Lazio.

**Informed Consent Statement:** Informed consent was obtained from all subjects involved in the study.

**Data Availability Statement:** The raw data supporting the conclusions of this article will be made available by the authors on request.

**Conflicts of Interest:** The authors declare no conflicts of interest.

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
