# Peer review of "Reflections on Recidivism and Relapse Prevention among Italian Justice-Involved Juveniles: A General Overview"

_socsci, doi:10.3390/socsci13050254_

Round 1

Reviewer 1 Report

Comments and Suggestions for Authors

I have a number of problems and issues with this paper, beginning with the research design, which is cross-sectional and descriptive (as opposed to comparative or exploratory), and is thus the weakest form of research design.  But even within the limits of this design, the authors have not pursued further analyses of their data that would have made the paper much more of a contribution to the literature on youth offending.  Before addressing some specific concerns in that regard, I find it interesting that the authors make several observations in introducing the paper that allude to its potential significance.  For example, they write that “recidivism is widespread and poorly studied”!  One could argue both for and against this proposition, but of relevance here is that this paper does not itself study recidivism, and is thus irrelevant to the proposition.  Likewise, the authors point to the importance of investigating “the meaning of criminal conduct of …juveniles”!  Here again, this paper does no such thing!

With respect to the data and analyses, the authors have collected self-reported delinquency data, neighborhood perceptions, and risk assessments, but present these in only descriptive form.  That means there is no effort at looking for correlations, for example, despite there being multiple opportunities to do so:  Do the risk factors have any correlation with the self-reports, e.g., higher risk youth engage in more and more serious delinquency?    Do older youth have more and more serious delinquency?  What about males compared with females?  And Italians compared with foreign nationals? Do the foreign nationals have different neighborhood perceptions than Italians?  Etc., etc.  Interestingly, if in the self-reports of delinquency, the authors had asked how many times reported offenses had been committed, they would have had an indicator of recidivism! But this was not done. 

Finally, there are a few statements in the paper that are not clear on their face.  For example, I do not know what “excessive security of not committing more crimes” means?  Nor, what it means that not all juveniles allocated to the 17 centers “adhered” to the study and were thus not included?

Comments on the Quality of English Language

See above.

Reviewer 2 Report

Comments and Suggestions for Authors

I am not sure of the hypothesis of or the motivation for this research.  It could be meant to replicate research identifying risk and protective factors.  But it is ill-equipped to do so, because it looks only at known offenders.  Thus it only investigates the cause (e.g., bad home life) of an effect (crime), not the effect of a cause.  To study the latter, research on the prevalence of identified "risk factors" in the non-offending population would need to be carried out.  Additionally, apparently the information gathered was entirely through self-report, so its accuracy is not clear. 

Perhaps the goal of the authors was simply to describe the risk factors other research has discovered and report the prevalence of those factors in the sample.  If so, that should be made very clear.  For instance, why does Table 1 only include 6 risk factors?  Were these taken from research?  Were they the top 6 risk factors identified in the research?  Why weren't the factors in the other tables included in Table 1, or at least mentioned as factors to be considered later?  And so on.

It would also be useful to know what the authors think about the fact that, in Tables 1 and Tables 3 through 8 no single risk/protective factor was found in more than 50% of the sample, and most were found in a much smaller proportion of the sample.  Is this consistent with other research?  Why or why not? 

Some of the authors' interpretations were confusing: for instance, they seem to think the fact that "only" 52% of their sample does not want to return home is a positive; they emphasize the presence of guns in the neighborhood but not of gangs/"contact with others who had involve[ment] in criminality"; on p. 17, they seem to conclude that less "psychological support" leads to more stress and lower likelihood of reintegration without explaining what they mean or how they arrived at their conclusion. 

The discussion of treatment programs is nice, but it is not clear what role it plays in this paper, or how it is tied to their research. 

In Tables 1-7, why not rank order the entries, to make it easier on the reader?   

Comments on the Quality of English Language

Some passages were quite well-written, others were not.  I've highlighted a few that need to be rewritten.

Round 2

Reviewer 1 Report

Comments and Suggestions for Authors

As noted previously, the phrase "excessive security of not committing more crimes" still appears on pages 7 and 12?  I think what might be actually meant is that some juveniles are overly confident in their ability to avoid further criminal behavior?  If so, this point seems to overlap with other items already included.

Comments on the Quality of English Language

No other comments!

Reviewer 2 Report

Comments and Suggestions for Authors

While the paper is much improved, the aim of the study is still not clear, perhaps because the English is hard to understand.     

Comments on the Quality of English Language

Not good.
